# Bifidobacterial Transfer from Mother to Child as Examined by an Animal Model

**DOI:** 10.3390/microorganisms7090293

**Published:** 2019-08-27

**Authors:** Walter Mancino, Sabrina Duranti, Leonardo Mancabelli, Giulia Longhi, Rosaria Anzalone, Christian Milani, Gabriele Andrea Lugli, Luca Carnevali, Rosario Statello, Andrea Sgoifo, Douwe van Sinderen, Marco Ventura, Francesca Turroni

**Affiliations:** 1Laboratory of Probiogenomics, Department of Chemistry, Life Sciences and Environmental Sustainability, University of Parma, 43124 Parma, Italy; 2GenProbio srl, 43124 Parma, Italy; 3Stress Physiology Laboratory, Department of Chemistry, Life Sciences and Environmental Sustainability, University of Parma, 43124 Parma, Italy; 4Microbiome Research Hub, University of Parma, 43124 Parma, Italy; 5School of Microbiology & APC Microbiome Institute, University College Cork, T12 YT20 Cork, Ireland

**Keywords:** bifidobacteria, infant gut microbiota, microbe-host coevolution, metagenomics

## Abstract

Bifidobacteria commonly constitute the most abundant group of microorganisms in the healthy infant gut. Their intestinal establishment is believed to be maternally driven, and their acquisition has even been postulated to occur during pregnancy. In the current study, we evaluated bifidobacterial mother-to infant transmission events in a rat model by means of quantitative PCR (qPCR), as well as by Internally Transcribed Spacer (ITS) bifidobacterial profiling. The occurrence of strains supplied by mothers during pregnancy to their corresponding newborns was observed and identified by analysis immediately following C-section delivery. These findings provide intriguing support for the existence of an unknown route to facilitate bifidobacterial transfer during the very early stages of life.

## 1. Introduction

Bifidobacteria are dominant members of the infant gut microbiota, although their numbers are known to decrease during weaning and adolescence, with a further decline in relative abundance in the elderly population [1,2]. Their role as natural modulators of the host immune system, as well as their impact on the physiology and metabolism of the human large intestine are subject to significant scientific scrutiny [1,3]. Due to these purported health benefits it is no surprise that bifidobacteria are frequently incorporated as functional ingredients in various (functional) foods and pharmaceutical products. Bacterial colonization of the human gut represents a very active field of fundamental and applied research, where bifidobacteria represent a key target group, and where strategies to influence the development, composition, and activities of the infant bifidobacterial members of the gut microbiota by means of nutraceutical products (e.g., probiotics and/or prebiotics) are very topical.

Various publications have described that the first microbial consortia in the gut of newborns may harbor bacteria that are vertically transmitted from the mother [4,5,6,7]. In this context, it has been discovered that amongst these maternally inherited bacteria, bifidobacteria represent an important group of gut commensals [8,9,10]. A recent study focusing on the identification of vertically acquired bifidobacterial strains based on a combination of shotgun metagenomics data, Internally Transcribed Spacer (ITS) bifidobacterial profiling analysis and a cultivation approach, described the finding of several bifidobacterial strains, including members of the *Bifidobacterium breve, Bifidobacterium longum* subsp. *longum* and *Bifidobacterium bifidum* species, which were shared between mother-infant dyads [9]. The identification of vertically transmitted bifidobacterial strains represents a key example of microbe-host co-evolution, and is a phenomenon that was also identified in other primate and non-primate mammals, where bifidobacteria were found to be commonly inherited from mother to her offspring, and where mother’s milk appears to represent an important route to facilitate such transfer events [11]. Furthermore, a recent study revealed that the composition of human milk is able to influence the gut microbiota of newborns, with potential implications for infant development and health [12].

The heretofore accepted dogma that the infant gut is sterile before birth has in recent years been challenged by different studies which suggest the existence of an intra-uterine and placenta bacterial community [13,14,15], though the presence and role of a genuine pre-birth microbiota is still controversial and subject to debate [14,16,17].

In the current study, we investigated vertical transmission events of bifidobacteria employing an in vivo model (*Rattus norvegicus*), resulting in a different bifidobacterial inheritance outcome when pregnant animals were treated with a single *Bifidobacterium* strain as compared to a mix of bifidobacterial strains belonging to three different *Bifidobacterium* species, i.e., *B. bifidum*, *B. breve* and *B. longum*.

## 2. Results and Discussion

### 2.1. Evaluation of Vertical Transmission of Bifidobacteria under in Vivo Conditions 

As previously shown by metagenomic attempts, identical bifidobacterial strains have been found in the fecal samples of mother-newborn dyads [7,9]. In order to evaluate possible microbial transfer by pregnant rats treated with a bifidobacterial strain mix consisting of *B. bifidum* PRL2010, *B. breve* 1895B and *B. longum* subsp. *longum* 1886 strains, or rats treated with *B. bifidum* strain PRL2010 only, animals were divided in two different groups. The first group consisted of three female rats, which were treated with the bifidobacterial mix (Mix Colonization Group—MCG), whereas the second group encompassed nine female rats, to which only PRL2010 was administered (PRL2010 Group—PG). Notably, the bifidobacterial strains used were, previously, isolated from fecal samples from healthy breast-fed infants [11,18,19]. However, even if these bacterial strains were not belonging to the indigenous microbiota of rat, when administered to these animals during pregnancy, they are able to colonize the gut of rat (see below). A similar result has been described previously by Jimenez et al., where a human infant gut commensal, *Enterococcus faecium*, was able to be vertically transmitted in pregnant mice. 

At the start of the experiment, rats were checked for the presence of strains PRL2010, 1895B and/or 1886B in fecal samples by means of PCR using strain-specific primer pairs, revealing that, as expected, these bifidobacterial strains were absent in the animals enrolled in this study and prior to them being fed any of the strains. Dams were administered a daily dose of 10^9^ colony forming units (CFU) of *B. bifidum* PRL2010 or a mix of *B. bifidum* PRL2010, *B. longum* subsp. *longum* 1886B and *B. breve* 1895B for 21 days. Presence and clearance of *B. bifidum* PRL2010, *B. longum* subsp. *longum* 1886B and *B. breve* 1895B were monitored during the gestation period using a qPCR approach based on strain-specific primers (Figure 1). Following Caesarian delivery of the pups, rats were sacrificed and their caecum was removed and assayed for the presence of PRL2010, 1886B and 1895B by means of qPCR. Interestingly, the microbial density estimated by qPCR of *B. bifidum* PRL2010 in the caecal samples of mothers ranged from 10^4^ to 10^7^ CFU, whereas in caecal samples of pups, the abundance ranged from 10^3^ to 10^6^ CFU (Figure 1). Similarly, the qPCR estimated abundance of *B. breve* 1895B in the caecum of mothers and of newborns ranged from 10^4^ to 10^7^ and 10^2^ to 10^6^, respectively (Figure 1). The cell load determined by qPCR of *B. longum* subsp. *longum* 1886B in the mother’s gut and in pup’s caecum varies from 10^3^ to 10^7^ and 10^2^ to 10^4^, respectively (Figure 1). 

In addition, we isolated viable PRL2010/1895B/1897B cells from the caecal samples of the mothers by means of direct cultivation of caecal contents of animals on mMRS followed by the precise strain-assignment of the isolated cells using a PCR approach based on strain-specific primers.

### 2.2. Maternal Inheritance of B. bifidum PRL2010 Strain 

Since *B. bifidum* strain PRL2010 has been shown to represent a model infant gut commensal [9,18,20,21,22,23,24,25,26,27,28,29,30,31,32,33,34], and as described above for the Mix Colonization Group—MCG, this strain displayed the highest level of vertical transmission from mother to newborns, we decided to further examine the maternal inheritance of strain PRL2010. Bifidobacterial inheritance of nine pregnant rats receiving a daily dose of 10^9^ CFU of *B. bifidum* PRL2010 for three weeks was monitored by qPCR (Figure 2). These rats delivered their puppies by Caesarian section and PRL2010 levels were evaluated by qPCR using strain-specific primers targeting both the caecal sample of the mother and the corresponding pups (Figure 2). The estimated abundance of PRL2010 cells by qPCR ranged from 10^3^ to 10^4^ CFU and from 10^2^ to 10^4^ CFU in mother and newborns samples, respectively (Figure 2). Notably, the PRL2010 colonization was lower in PG mothers respect to MCG mothers (*p* value < 0.001). Concurrently, the genome copy number of PRL2010 in the newborns from PG group was lower respect those observed in MCG newborns group (*p* value < 0.001). These findings suggest that microbe-microbe interactions provide an advantage in the vertical transmission efficiency of these species in the infant gut. In addition, such results corroborate previous data regarding the existence of syntrophic interactions between bifidobacteria in the infant gut [9,11,33]. 

However, the bacterial load of PRL2010 in the MCG newborns group resulted lower respect to their mothers. In contrast, the PG newborns group showed a higher abundance of PRL2010 cells when compared to their mothers (Appendix A). Probably, the load of *B. bifidum* PRL2010 cells in the newborns of PG group is higher respect to their mothers because it has been administered as a mono-strain supplement. Conversely, the simultaneously supplementation of bifidobacterial mix encompassing different strains in MCG group contributed to the decreased level of PRL2010 cells in the newborns.

### 2.3. Identification of DNA Belonging to PRL2010 in Different Rat Body Sites 

Bifidobacterial DNA occurrence in other body sites of rats such as placenta and blood samples, which was collected from mothers immediately after Caesarian delivery, was investigated by qPCR using PRL2010-specific primers. Remarkably, these experiments revealed the presence of DNA belonging to PRL2010 in the placental tissue but not in blood (Figure 3). Similar results were described previously by Jimenez et al. in a trial examining the vertical transmission of *E. faecium* HA1 in pregnant mice [35]. Any attempts directed to cultivate PRL2010 cells from placenta samples under mMRS did not result in the isolation of viable cells. These findings suggest that either the rat placenta can only be reached by DNA from lysed PRL2010 cells or that this body compartment only contains dormant PRL2010 cells. These data may also open a novel and intriguing scenario of prenatal colonization of rats by *B. bifidum* PRL2010.

### 2.4. ITS Bifidobacterial Profiling of the Caecum of Mothers and Newborns 

In order to further characterize the bifidobacterial composition of caecal samples from mothers and puppies, ITS bifidobacterial profiling analyses were performed [36] on these 30 samples, producing a total of 212,034 reads with an average of 7068 ± 6457 reads per sample (Table 1). These raw data were processed to identify and classify reads into clusters of identical sequences (OTUs). In detail, we focused our interest on OTUs belonging to the species *B. bifidum*, *B. breve* and *B*. *longum* subsp. *longum*. Interestingly, all puppies of the MCG group shared these three OTUs with the corresponding mothers’ samples (Figure 4). Moreover, a comparison between PG mothers with their corresponding puppies showed that in 70% of cases the OTUs belonging to the species *B. bifidum*, *B. breve* and *B*. *longum* subsp. *longum* are shared (Figure 4). In order to confirm the presence of *B. bifidum*, *B. breve* and *B*. *longum* subsp. *longum* in PG group, we performed a qPCR using species-specific primers using mothers’ caeca (Figure 4). These data revealed that *B. breve* and *B*. *longum* subsp. *longum* species were present in the gut microbiota of pregnant rats, even if they were not supplemented to the animals (Figure 4). 

These results further support the notion of transfer of (bifido)bacterial DNA and/or cells between mother and puppies [11,37] and reinforced the earlier finding that co-existence of multiple bifidobacterial species allows a more efficient transmission of (bifido)bacterial DNA and/or cells from mothers to their newborns [9,11,33]. 

## 3. Materials and Methods

### 3.1. Experiment Design and Bifidobacterial Treatment of Rats 

Experiments were performed in accordance with the European Community Council Directive 2010/63/UE and approved by the Italian legislation on animal experimentation (D.L. 04/04/2014, n. 26, authorization no. 370/2018-PR). All efforts were made to reduce sample size and minimize animal suffering. For the purposes of the current study we employed adult (4–6 months) Wistar rats (*Rattus norvegicus*) which were bred in house under standard conditions [38,39]. The timeline of animal treatment procedures is schematically depicted in Appendix A. The duration of the animal trial was four weeks in total. Mating was allowed during week 1 and for this purpose, 12 adult female rats (*Rattus norvegicus*) were coupled with 12 male rats. Pairs were kept in different cages in rooms with controlled temperature (22 ± 2 °C) and humidity (60 ± 10%) and maintained in a 12/12 light/dark cycle (light on from 19: 00 to 7: 00 h), with food and water *ad libitum*. Following the one week mating period, all female rats were separated and kept in individual cages and were every day orally treated with bifidobacterial strains for three weeks, which represented the gestation period. One group of three female rats (which are here referred to as W1, W2 and W3) was orally inoculated with a mix of three different strains, i.e., *B. bifidum* PRL2010, *B. breve* 1895B and *B. longum* subsp. *longum* 1886B (Mix Colonization Group–MCG). The second group with the remaining nine female rats (W4, W5, W6, W7, W8, W9, W10, W11 and W12) was orally treated with *B. bifidum* PRL2010 only (PRL2010 Group—PG). All the strains used in this study were previously isolated from infant stool samples [18,19]. In order to evaluate bifidobacterial transfer, fecal samples were collected at five different time points. The first time point was before the oral administration of bifidobacteria (T0). Then, we collected fecal samples at four time points, i.e., at five, eight, 12 and 17 days (T1, T2, T3 and T4) (Appendix A). On the 19th day of bifidobacterial treatment (i.e., on the day of birth, but prior to labor) female animals were anesthetized with isoflurane (2% in 100% oxygen) and Caesarian delivery was performed under a laminar flow hood and all the surgical instruments used were previously sterilized in order to prevent any microbial contamination. Placentas were harvested and caecum and blood samples were collected from dams and pups.

### 3.2. Bifidobacterium Strains Growth Conditions

All strains used in this study were cultivated in an anaerobic atmosphere (2.99% H_2_, 17.01% CO_2_ and 80% N_2_) in an anaerobic chamber (Concept 400, Ruskin) on De Man-Rogosa-Sharp (MRS) broth (Scharlau Chemie, Barcelona, Spain) supplemented with 0.05% (*w/v*) L-cysteine hydrochloride (Sigma-Aldrich, Saint Louis, USA) and incubated at 37 °C. Microbial cultures were harvested by centrifugation (3000 rpm for 8 min), washed and resuspended in 500 µL of 2 % (*w/v*) sucrose solution. The viable count of each inoculum was determined by retrospective plating on MRS.

### 3.3. DNA Extraction and qPCR

Bacterial DNA extraction from fecal samples was performed following the manufacturer’s protocol of the QIAamp Fast DNA stool Mini Kit (Qiagen Ltd., Strasse, Germany). Furthermore, bifidobacterial DNA presence was evaluated for the mother’s caecum, placentas, blood samples, and from puppies’ caecal samples. Specifically, the bacterial DNA from puppies and mothers’ caecum and from placentas was extracted following the protocol of Power Viral Environmental RNA/DNA Isolation Kit (Mo Bio, Hilden, Germany), whereas the bacterial DNA from mother’s blood was extracted following the protocol of the DNeasy Blood and Tissue Kit (Qiagen Ltd., Strasse, Germany). 

Quantitative PCR (qPCR) was performed as described previously [32]. For *B. bifidum* PRL2010 were used primers Bbif_0282Fw (5′-GCGAACAATGATGGCACCTA-3′) and Bbif_0282Rv (5′-GTCGAACACCACGACGATGT-3′) [24], for *B. breve* 1895B were used primers BBR7E_0534_fw (5′-AGCGACGATATGATGCAATG-3′) and BBR7E_0534_rev (5′-CGTGAATACGCTGCACAGTC-3′) and for *B. longum* subsp. *longum* 1886B were used primers B1886_0443_fw (5′-AAGCCAAGGACATGTTCGAC-3′) and B1886_0443_rev (5′-TGGTGTATCTGGCGTTCTTG-3′) [19]. For species specific qPCR were used following primer pairs: Bbif1 (5′-CCACATGATCGCATGTGATTG-3′) and Bbif2 (5′-CCGAAGGCTTGCTCCCAAA-3′) for *B. bifidum* species [29], Bbre1 (5′-CCGGATGCTCCATCACAC-3′) and Bbre2 (5′-ACAAAGTGCCTTGCTCCCT-3′) for *B.breve* species [40], Blon1 (5′-TTCCAGTTGATCGCATGGTC-3′) and Blon2 (5′-GGGAAGCCGTATCTCTACGA-3′) for *B. longum*.

### 3.4. Bifidobacterium Strain Isolation From Mothers’ Caecum

The collected caecum of the mothers was homogenized and serial dilutions (1:10) were performed. All dilutions were cultivated on MRS agar (Scharlau Chemie, Barcelona, Spain) supplemented with 0.05% (*w/v*) L-cysteine hydrochloride (Sigma-Aldrich) and 50 μg/mL mupirocin (Delchimica, Napoli, Italy). After 48h of incubation in an anaerobic atmosphere (2.99% H_2_, 17.01% CO_2_ and 80% N_2_) in a chamber (Concept 400, Ruskin), morphologically distinct colonies were selected and cultivated in MRS broth (Scharlau Chemie, Barcelona, Spain) supplemented with 0.05% (*w/v*) L-cysteine hydrochloride (Sigma-Aldrich). Subsequently, after overnight growth, bacterial DNA was extracted as described previously [41]. The identification of specific strains was obtained using a PCR approach based on strain-specific primers (see above). 

### 3.5. Bifidobacterial ITS PCR Amplification and Sequencing

Following bacterial DNA extraction from caecal samples of mothers and puppies, partial ITS sequences were amplified using primer pair Probio-bif_Uni/Probio-bif_Rev, which targets the spacer region between the 16S rRNA and 23S rRNA genes within the ribosomal RNA (rRNA) locus [36]. At the same time, we prepared a mock community (Mock Bifidobacterial Community), consisting of a pool of known concentrations of 11 different *Bifidobacterium* strains prepared by combining equal concentration of bacterial DNA. The DNA from the mix was diluted to produce a final DNA concentration of 2 ng/μL, and 4 μL of these dilutions were used in each PCR reaction using primer pair Probio-bif_Uni/Probio-bif_Rev. Illumina adapter overhang nucleotide sequences were added to the partial ITS amplicons, which were further processed employing the 16S Metagenomic Sequencing Library Preparation Protocol (Part #15044223 Rev. B–Illumina). The library preparation was performed as described above for the 16S rRNA microbial profiling analyses.

Following sequencing, the.fastq files were processed using a custom script based on the QIIME software suite [42]. In order to reconstruct the complete Probio-bif_Uni / Probio-bif_Rev amplicons, the paired-end read pairs were assembled. Quality control retained sequences with a length between 100 and 400 bp and mean sequence quality score of >20, while sequences with homopolymers >7 bp in length and mismatched primers were removed. ITS Operational Taxonomic Units (OTUs) were defined at 100% sequence homology using uclust software [43]. All reads were classified to the lowest possible taxonomic rank using QIIME2 [42,44] and a reference dataset, consisting of an updated version of the bifidobacterial ITS database [36].

### 3.6. Data Deposition

Raw sequences of the bifidobacterial ITS profiling experiments are accessible through SRA study accession numbers PRJNA556137.

## 4. Conclusions

This study was aimed to investigate the DNA bifidobacterial transmission from female rats to their corresponding pups through a vertical route. Notably, the efficiency/yield of these DNA transmission events may be influenced by the co-presence of different bifidobacterial strains. The above results confirm previous genus-level overviews [37] and underpin the notion that bifidobacterial transmission is characterized by the development of extensive microbe-microbe co-operation [7,11,33]. In fact, our data suggest that bifidobacterial communities have established co-operative behavior between co-colonizers, acting as evolutionary drivers in the mammalian gut microbiota. This assumption is further reinforced by previous studies that have described the occurrence of cross-feeding between specific bifidobacterial species [7,33]. In addition, our findings have provided preliminary insights about the existence of possible bifidobacterial colonization at the preterm level. Nevertheless, the lack of any data about the viability of PRL2010 cells cannot fully support the existence of fetal colonization of bifidobacteria, and further experiments will be needed in order to investigate this intriguing route of bacterial colonization.

## Figures and Tables

**Figure 1 microorganisms-07-00293-f001:**
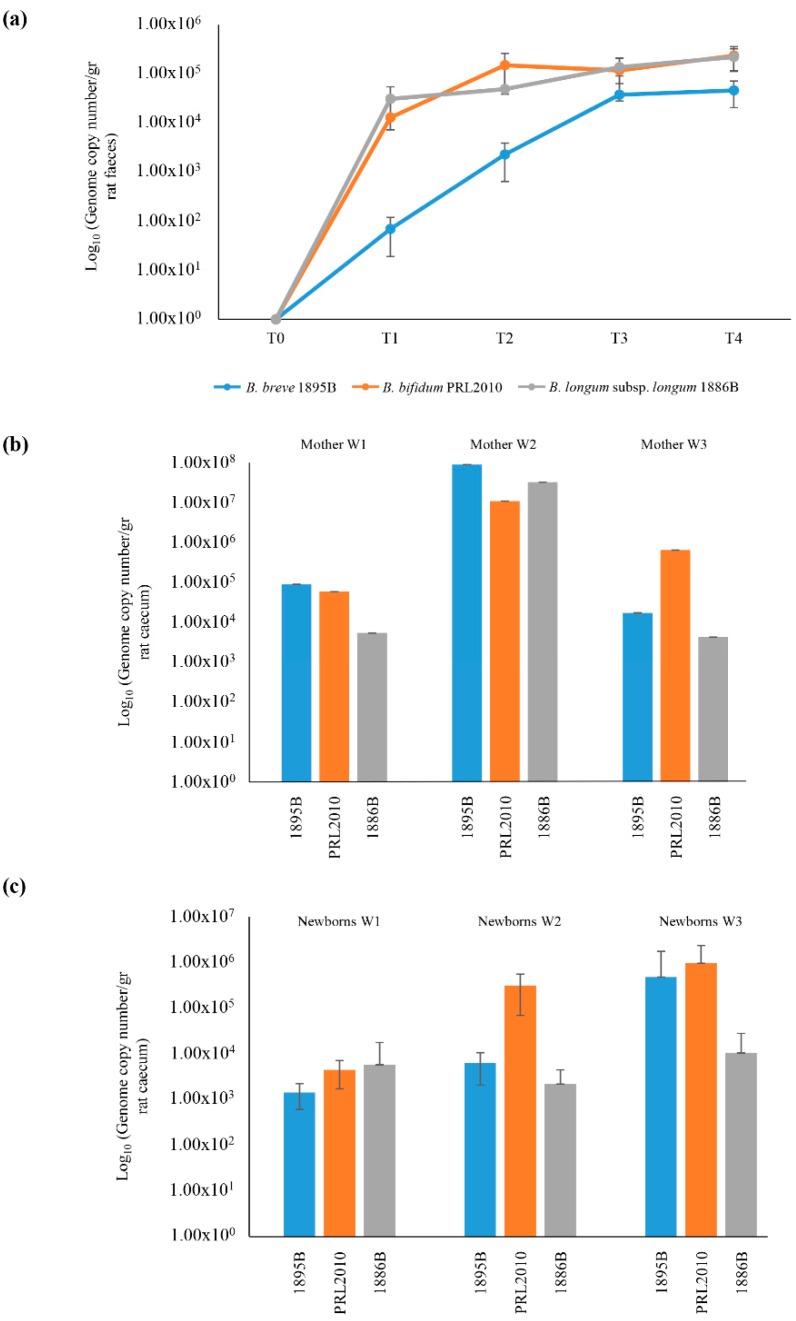
Schematic representation of the vertical transmission of the bifidobacterial mix in rat model treated with *B. breve* 1895B, *B. bifidum* PRL2010 and *B. longum* subsp. *longum* 1886B strains (Mix Colonization Group—MCG). Panel (**a**) shows the average of DNA presence of the three strains observed during the bifidobacterial administration. Each point represents the average of the log-population size ± standard deviation for three rats. Panel (**b**) displays the presence of *B. breve* 1895B, *B. bifidum* PRL2010 and *B. longum* subsp. *longum* 1886B in the caecum of female rats. Panel (**c**) exhibits bifidobacterial retrieval from puppies’ caecum. Each pillar represents the average presence for each *Bifidobacterium* strain ± standard deviation.

**Figure 2 microorganisms-07-00293-f002:**
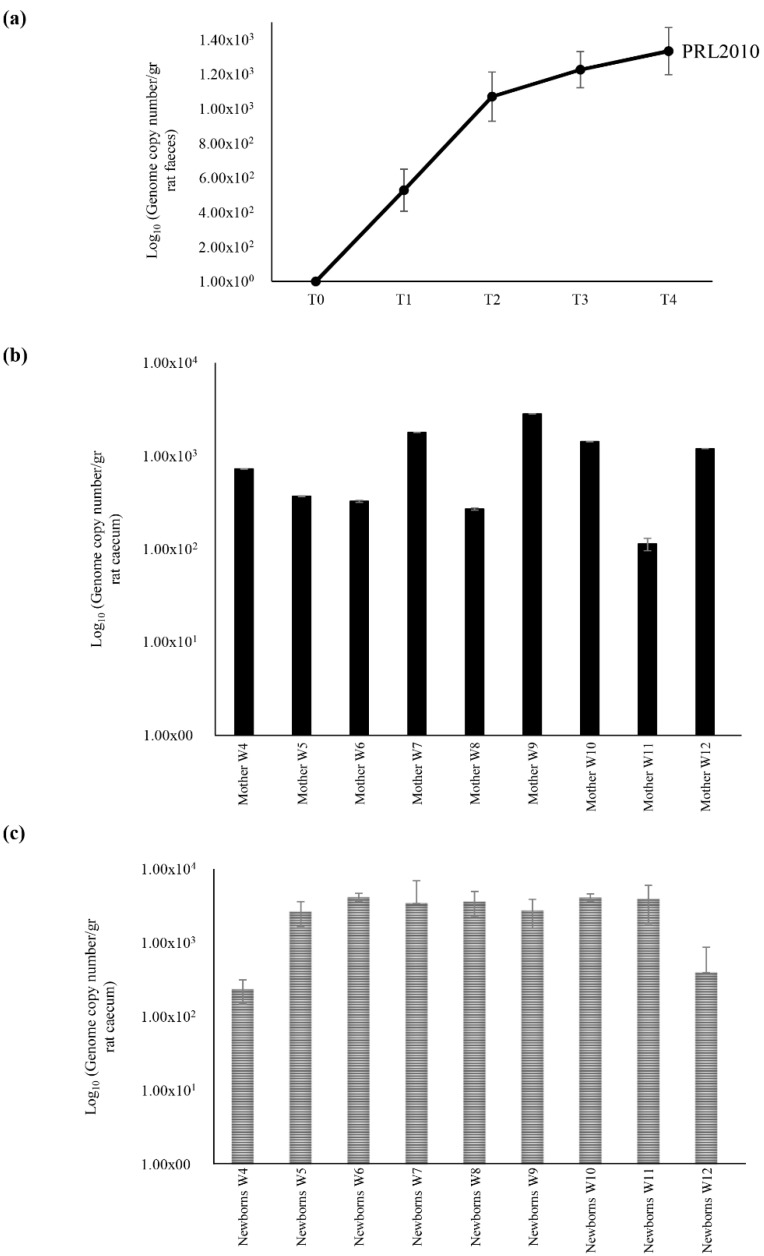
Schematic representation of vertical inheritance of *B. bifidum* strain PRL2010, when administered solely in a rat model (PRL2010 Group—PG). Panel (**a**) shows the population sizes of *B. bifidum* PRL2010 present in the intestine of female rats. Each point represents the average of the log-population size ± standard deviation for nine rats. Panel (**b**) displays the *B. bifidum* PRL2010 retrieval ± standard deviation for each female rat used. Panel (**c**) exhibits the presence of *B. bifidum* PRL2010 in the caecum from puppies. Each pillar represents the average colonization ± standard deviation of puppies for each female rat.

**Figure 3 microorganisms-07-00293-f003:**
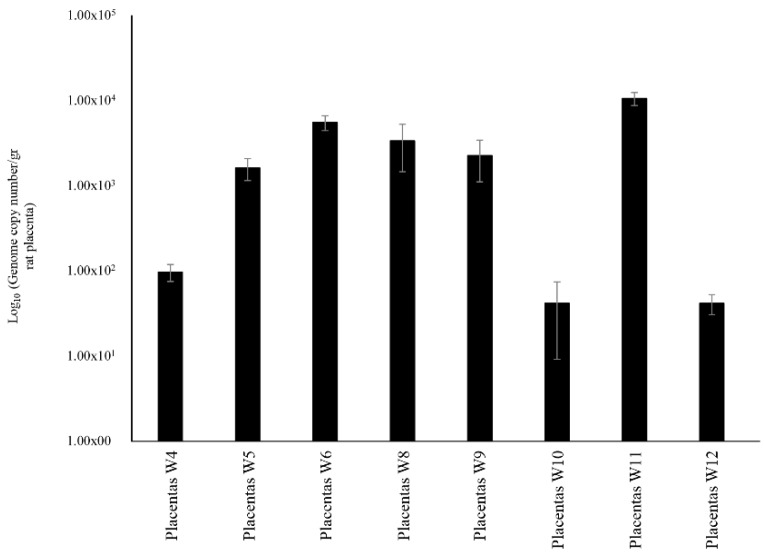
Schematic representation of the *B. bifidum* PRL2010 DNA load observed in placentas’ samples. This graphic displays the level of DNA belonging to *B. bifidum* PRL2010 present in the placenta of rats. Each pillar represents the average retrieval ± standard deviation of *B. bifidum* PRL2010 from the placenta of rats.

**Figure 4 microorganisms-07-00293-f004:**
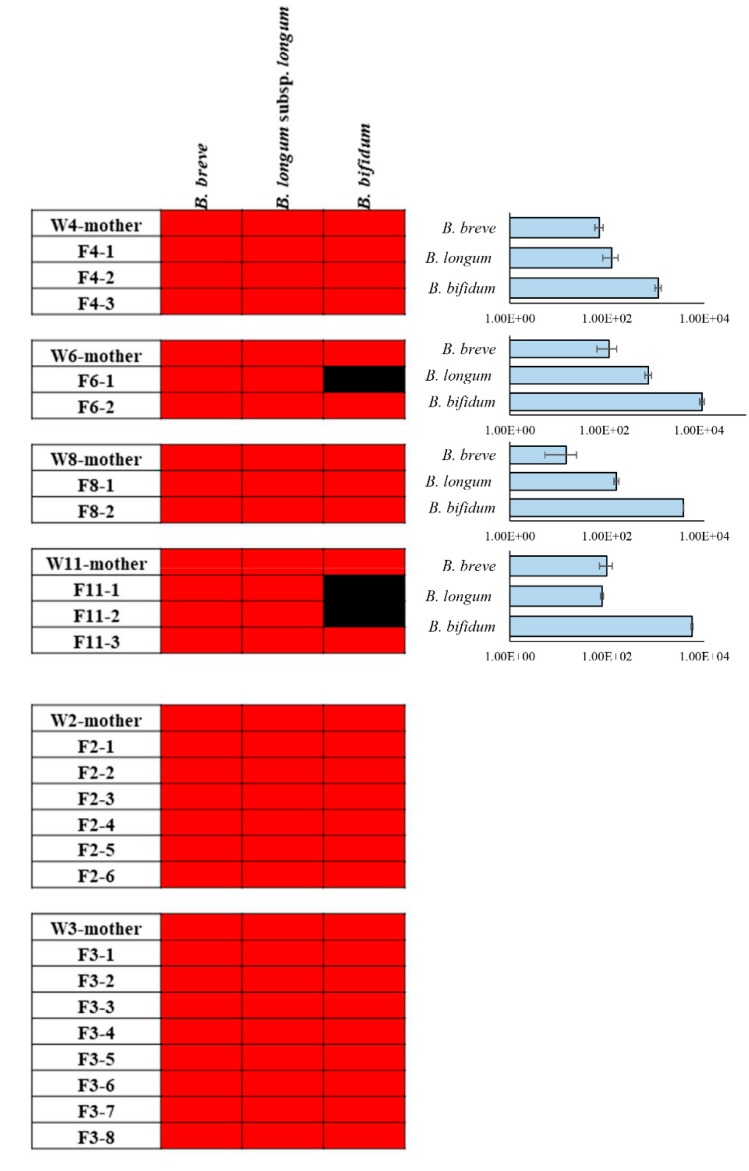
Heat map of bifidobacterial OTUs belonging to *B. bifidum*, *B. breve* and *B. longum* subsp. *longum*. Red shading represents presence, and black shading indicates absence. The pillars next to each heat map display the DNA load of these bifidobacterial species in mothers’ caeca of PG group. The *x* axes represent the genome copy number /gr of cecum samples.

**Table 1 microorganisms-07-00293-t001:** Filtering table of the analyzed caecal samples.

Experiment Groups	Samples Name	Input Reads	Final Reads
PG	W4-mother	3423	3126
PG	F4-1	2589	2386
PG	F4-2	8551	8012
PG	F4-3	4126	3776
PG	W6-mother	2078	1970
PG	F6-1	2164	2036
PG	F6-2	5050	4852
PG	W8-mother	1239	1201
PG	F8-1	6365	5797
PG	F8-2	4527	4050
PG	W11-mother	1009	942
PG	F11-1	4106	3908
PG	F11-2	2133	2036
PG	F11-3	1692	1642
MCG	W2-mother	22,528	22,026
MCG	F2-1	12,703	11,906
MCG	F2-2	32,065	29,485
MCG	F2-3	11,715	10,850
MCG	F2-4	17,106	15,929
MCG	F2-5	13,992	12,724
MCG	F2-6	10,339	9739
MCG	W3-mother	2686	2640
MCG	F3-1	4818	4620
MCG	F3-2	8970	8489
MCG	F3-3	5346	4987
MCG	F3-4	5664	5373
MCG	F3-5	2765	2584
MCG	F3-6	12,523	11,182
MCG	F3-7	6129	5701
MCG	F3-8	8524	8065

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
