# Peer review of "Bifidobacterial Transfer from Mother to Child as Examined by an Animal Model"

_microorganisms, 2019, doi:10.3390/microorganisms7090293_

Round 1
Reviewer 1 Report
The main subject of this research work is the bacteria vertical transmission in maternity. The concept is very controversial and unclear yet. Nevertheless, I strongly agree that tracing commensal bacteria transfer from mother to child is important to extend our knowledge for host-microbe interaction and coevolution. In particular, in vivo rat model approach is essential and valuable to overcome the limitations of previous researches, and bifidobacteria is surely a primary target for this subject. Authors had already been contributed to finding significant clues in the human case for this issue using NGS analysis (Milani et al., 2015; Duranti et al., 2017). In this study, also, the ITS amplicon NGS and species-specific qPCR is used for tracing bifidobacteria deliver in female rat and their pups. And selected bifidobacteria strains were administrated to evaluate its vertical transmission. Unfortunately, however, the results and conclusions are partially confusing and unconvincing.
First, the results did not establish the route of vertical transmission in the qPCR approach and did not present direct evidence of delivering. Bacterial DNA was not found in the blood (L197). Placenta included bacterial DNA (Fig.3), but bacteria cultivation was failed (L199). For reference, Jimenez, E. et al. (Curr. Microbiol. 51, 270–274 (2005)) already reported similar research using labeled bacteria within a murine model.
Second, MCG and PG administration and its caecum qPCR results are only present a genome copy number increasing in the MCG case compared to the PG case. From this, authors concluded that ‘microbe-microbe interactions provide an advantage in the vertical transmission efficiency of these species in the infant gut’ (L184). I agree on it as a reasonable argument, however, cannot find linkage with the data. This animal model is not the germ-free condition, there is not any reaction information when the three strains are mixed, and even the caecum ITS result are not matched which is PG mothers (W4, W6, W8, and W11) also have three bifidobacteria strains (Fig. 4).
There is one more confusing point. In the Fig.2, authors did not refer to the strong increase of PRL2010 genome copy in the newborn caecum compared to their mother (L181). Just mentioned as ‘qPCR ranged from 103 to 104 CFU and from 102 to 104 CFU in mother and newborns. But by simple calculation, newborns show about -2 to +60 fold change compared to each mother. It is a totally different viewpoint for information and tendency. Summarized:
Genome copy in the caecum: MCG mothers > PG mothers Genome copy in the caecum: MCG newborns > PG newborns Genome copy in the caecum: MCG mother > MCG newborns Genome copy in the caecum: PG mother < PG newborns
Although the purpose of this paper is providing preliminary insights, considering the importance and controversy of this topic, much prudence, refinedness, and clear focus are required.
Author Response
We have addressed all comments and suggestions made by the expert referees and have implemented all the proposed changes. All comments were considered very helpful and valuable, significantly improving the manuscript in its overall quality.
In the following, we discuss the individual suggestions/comments made by the reviewer in detail:
Athour: we have rephrased the sentence including Jimenez, E. et al. reference (L214-216). In accordance with this reviewer, the model that we have used is a conventional rat (not germ-free animal), which implies that the mothers could be already naturally colonized by bifidobacteria. In order to evaluate the presence of breve and B. longum subsp. longum species in the mothers’ caeca of the PG group we performed a qPCR analysis (L232-235, L241-242 and Fig. 4 modified). As suggested by this reviewer, we have modified the text. Furthermore, as required by the reviewer, we have performed an additional analysis, displaying the PRL2010 load of mothers PG vs. newborn PG and the PRL2010 load of mothers MCG vs. newborn MCG (Fig. S2, L184-201 and L259). As indicated by this referee we modified the conclusion section by reducing the emphasis placed about the prenatal colonization/vertical transmission of bifidobacteria before the birth (L248-260).
Reviewer 2 Report
This is an study that try to contribute to support the existence of an unknown rute of bacterial transfer during the early stages of life. The origin of milk microbiota by a entero-mamary translocation is a more or less recent and attractive idea that unfortunatelly is still difficult to demostrate.
Please, add information about the origin of the Bifidobacterium strains studied.
Are they of rat origin? Are vertical transmission possible using strains from a different origin that the animal model?
Is it possible to perform a caesarian delivery under strict sterile conditions?
Since all data are supported by molecular results, I agree that the conclusion must be that the lack of data cannot support the existence of fetal colonization of bifidobacterial.
Please check the paper: Composition and Variation of the human milk microbiota are influenced by maternal and early-life factors by Moossavi et al. 2019
Author Response
We have addressed all comments and suggestions made by the expert referees and have implemented all the proposed changes. All comments were considered very helpful and valuable, significantly improving the manuscript in its overall quality.
In the following, we discuss the individual suggestions/comments made by the reviewer in detail:
Reviewer-2:This is an study that try to contribute to support the existence of an unknown rute of bacterial transfer during the early stages of life. The origin of milk microbiota by a entero-mamary translocation is a more or less recent and attractive idea that unfortunatelly is still difficult to demostrate.
Please, add information about the origin of the Bifidobacterium strains studied.
Are they of rat origin? Are vertical transmission possible using strains from a different origin that the animal model?
Author: As indicated by this reviewer, we have provided the information about the ecological origin of bifidobacterial strains used in this study (L77-78). Bifidobacterial strains used in this study were isolated from infant stool samples (lines 148-149 and lines 77-78) as described previously (Turroni et al., 2010, Proc Natl Acad Sci; Duranti et al., 2017, Microbiome). In addition, we have explained how these bifidobacterial strains even if they are not from rat origin they were able to be vertically transmitted from the mother to the child (L152-157).
Reviewer-2: Is it possible to perform a caesarian delivery under strict sterile conditions?
Since all data are supported by molecular results, I agree that the conclusion must be that the lack of data cannot support the existence of fetal colonization of bifidobacterial.
Please check the paper: Composition and Variation of the human milk microbiota are influenced by maternal and early-life factors by Moossavi et al. 2019
Author: In accordance with this referee, we have added details regarding the strictly sterile conditions followed during the Caesarian delivery (L83-84). We agree with this reviewer and we have clearly indicated this comment in the text (L232-235). As suggested by this referee, we have mentioned the study of Moossavi et al., 2019, Cell (L48-50).
Round 2
Reviewer 1 Report
Thank you for your interesting work and active revision.
I can see some minor spell error. It should be changed in the final version.
I hope that our knowledge of bacterial vertical transmission will be updated more with your contribution.
Author Response
In the following, we discuss the individual suggestions/comments made by the reviewer 1 in detail:
Reviewer 1:
In accordance with this reviewer, we have checked all the manuscript in order to correct all some minor spell errors.
We corrected further typos so that the final draft of our manuscript in our opinion is now of a much better quality. Therefore, we hope that the revised contributions have addressed all reviewers and Academic Editors’ comments. Thank you very much for consideration of this revised manuscript.
Sincerely yours
Francesca Turroni